# Effect of Polydextrose on the Growth of *Pediococcus pentosaceus* as Well as Lactic Acid and Bacteriocin-like Inhibitory Substances (BLIS) Production

**DOI:** 10.3390/microorganisms10101898

**Published:** 2022-09-24

**Authors:** Maria Carolina Wanderley Porto, Pamela Oliveira de Souza de Azevedo, Felipe Rebello Lourenço, Attilio Converti, Michele Vitolo, Ricardo Pinheiro de Souza Oliveira

**Affiliations:** 1Department of Biochemical and Pharmaceutical Technology, Faculty of Pharmaceutical Sciences, University of São Paulo, Av. Lineu Prestes 580, São Paulo 05508900, Brazil; 2Department of Pharmaceutical Sciences, Faculty of Pharmaceutical Sciences, University of São Paulo, Av. Lineu Prestes 580, São Paulo 05508900, Brazil; 3Department of Civil, Chemical and Environmental Engineering, University of Genoa, Pole of Chemical Engineering, via Opera Pia 15, 16145 Genoa, Italy

**Keywords:** *Pediococcus pentosaceus* ATCC 43200, bacteriocin-like inhibitory substance, polydextrose, pathogens microorganisms

## Abstract

*Pediococcus pentosaceus* was cultivated in MRS medium supplemented or not with polydextrose under different conditions in order to evaluate its effect on cell growth, lactic acid and bacteriocin-like inhibitory substance (BLIS) production. Independent variables were pH (4.0, 5.0, 6.0), rotational speed (50, 100, 150 rpm), polydextrose concentration (0.5, 1.0, 1.5%) and temperature (25, 30, 35 °C), while cell concentration and productivity after 24 h, maximum specific growth rate, specific rate of substrate (glucose) consumption, volumetric and specific lactic acid productivities, yields of biomass and lactic acid on consumed substrate were the dependent. The maximum cell concentration (10.24 ± 0.16 g_X_ L^−1^) and productivity (0.42 ± 0.01 g_X_ L^−1^ h^−1^) were achieved at pH 6.0, 35 °C, 150 rpm using 1.5% polydextrose, while the maximum specific growth rate (0.99 ± 0.01 h^−1^) and yield of biomass (2.96 ± 0.34 g_X_ g_S_^−1^) were achieved at the same pH and polydextrose concentration, but at 25 °C and 50 rpm. The specific substrate consumption rate (0.09 ± 0.02 g_S_ g_X_^−1^ h^−1^) and the volumetric lactic acid productivity (0.44 ± 0.02 g_P_ L^−1^ h^−1^) were maximized at pH 6.0, 35 °C, 50 rpm and 0.5% polydextrose. BLIS produced in this last run displayed the highest antibacterial activity against *Escherichia coli*, while the same activity was displayed against *Enterococcus faecium* using 1.5% polydextrose. These results appear to be quite promising in view of possible production of this BLIS as an antibacterial agent in the food industry.

## 1. Introduction

Polydextrose is a polymer composed of glucose units linked each to other through alpha and/or beta glycosidic bonds, mostly of the 1→6 type, thus leading to several aleatory molecular branches. Because of its highly branched structure, it is fermented slowly by a mix of bacteria in the distal portion of colon in humans and animals [1,2]. Properties of this polysaccharide are being investigated to discover possible effects on human gut microbiota and then to check whether it should be considered a prebiotic or not. Three criteria are usually taken into account to classify a substance as a prebiotic: (i) resistance to gastrointestinal activities under acidic environment including hydrolysis by mammalian enzymes and nutritional absorption; (ii) serving as a carbon source for intestinal bacterial strains, and (iii) acting as a substrate in fermentative processes, hence contributing to bacterial growth and the formation of beneficial products to the body [3,4]. Hooda et al. [5], who investigated polydextrose effects in healthy adult men consuming three polydextrose-containing snack bars per day (21 g day^−1^) for 21 days, observed that it stimulated the abundance of some intestinal bacteria belonging to Clostridiaceae, Veillonellaceae, *Faecalibacterium* sp., *Phascolarctobacterium* sp. and *Dialister* sp.

Regarding the substances released by polydextrose fermentation and their beneficial effects on health, Hernot et al. [6] reported that its fermentation in the colon reduces the production of short-chain fatty acids (acetate, propionate, and butyrate) and gases compared to other carbohydrates and oligosaccharides. Particularly, butyrate is produced after polydextrose consumption and is metabolized by colonocytes, enhancing the integrity of the colonic mucosa, and is behind protective and defense mechanisms [7]. In vitro experiments demonstrated that polydextrose fermentation reduced the levels of fecal ammonia, phenolic, indole and cresol compounds from protein fermentation [8], while enhancing the intestinal absorption of calcium, iron, and magnesium ions [9,10]. Moreover, food safety [7,11], capability of forming fatty acids [12], cholesterol reduction [1] and regulation of intestinal constipation [13] are some of the most reported and discussed clinical effects on humans.

There are many lactic-acid bacteria (LAB) that have been described as good bacteriocin producers [14,15], among which those belonging to the genus *Pediococcus* stand out. Some LAB strains are capable of producing antimicrobial compounds, such as organic acids, hydrogen peroxide, and bacteriocins or Bacteriocin-Like Inhibitory Substance (BLIS), which can inhibit the growth of spoilage and pathogenic bacteria [16,17]. BLIS are bacteriocins that are not obtained in pure form or fully characterized in terms of their amino acid sequences and molecular weight [18,19]. Bacteriocins and BLIS are peptides or proteins produced by ribosomes and secreted into the extracellular environment. They possess antimicrobial activity against pathogens, even at low concentrations, and are resistant to heat and pressure processes used in the food industry, allowing them to retain their activity during storage [20,21].

*Pediococcus pentosaceus* is a gram-positive, facultative anaerobe, catalase- and oxidase-negative bacterium, which is morphologically classified as a coccus commonly arranged in tetrads. Many strains of this species, which was successful in the fermentation of different matrices including goat milk [22], have been identified as producers of bacteriocins called pediocins [23], which are classified into the Class IIa of bacteriocins [24] and whose flexible structure is due to a hydrophilic N-terminal portion, characterized by the amino acid sequence YGNGV, and an amphiphilic or hydrophobic C-terminal. The applicability of pediocins, mainly PA-1 pediocin, is linked to their popularity as food preservatives able to minimize the growth of gram-positive bacteria such as *Listeria monocytogenes*, and their possible harmful effects to food consumers [25]. *P. pentosaceus* SM1, isolated from dry sausage, was found to coincide with *P. pentosaceus* 1934, the producer of a thermoresistant Class IIa bacteriocin with a molecular mass of 5.37 kD that proved to be active against *Micrococcus luteus* CECT 241, *Pediococcus acidilactici* ATCC 25740, *P. pentosaceus* ATCC 33316, *Lactococcus lactis* 11454, *Lactobacillus curvatus* ATCC 51436, *Lactobacillus sakei* CECT 906T, *Lactobacillus plantarum* NCAIM B 01133, NCAIM B 1074, CECT 220 and ATH. Despite its relatively low activity (70 AU mL^−1^), this bacteriocin, which was optimally produced under aerobic conditions, was stable for 1 h at 100–121 °C and for one year at −8 °C and −2 °C [26]. *P. pentosaceus* ST44M was isolated from Marula fruit (*Sclerocarya birrea*) and characterized by PCR analysis, 16SrDNA sequencing and typical carbohydrate reactions. The produced ST44M bacteriocin, with a molecular mass of about 6.5 kDa, was attributed to the Class IIa, was shown to be stable for 2 h at 100 °C and in a wide range of pH (2–12), and displayed antimicrobial activity against either gram-positive bacteria such as *Listeria ivanovii* subsp. *ivanovii* ATCC 19119 and *Enterococcus faecium* HKLHS or gram-negative bacteria (*E. coli* P8 and P40, *Pseudomonas aeruginosa* P22, *Pseudomonas* sp. P28 and *Klebsiella pneumoniae* P30). Its production was almost independent of temperature in the range 26–37 °C and appeared to be linked to the primary metabolism [27]. Two strains of *P. pentosaceus* (R38 and Cb4) isolated from vegetables were reported to produce thermostable bacteriocins strictly related to pediocin PA-1 that exerted antimicrobial activity against both gram-positive (*Listeria* sp.) and gram-negative (*Aeromonas* sp. and *Yersinia* sp.) indicator strains, the former being active also against *Staphylococcus aureus* [28]. Finally, *P. pentosaceus* ATCC 43200, which was isolated by Piva and Headon [29] from cucumber and firstly named *P. pentosaceus* FBB61, was proven to be a very good, commercially available pediocin or BLIS producer [30,31,32].

Despite the large use of prebiotics [33], there is a relative lack in the literature of studies on the influence of polydextrose on bacteriocin production [34]; therefore, the effect of this polysaccharide on the fermentation ability of *P. pentosaceus* ATCC 43200 has thoroughly been investigated in this study at different concentrations, aiming to improve BLIS production for possible use as a food biopreservative.

## 2. Material and Methods

### 2.1. Bacterial Strains and Growth Conditions

*P. pentosaceus* ATCC 43200 was grown in de Man, Rogosa, and Sharpe (MRS) broth (Difco, Detroit, MI, USA), some samples supplemented with polydextrose. Temperature, pH, polydextrose concentration and rotational speed were selected as the independent variables and set according to the 2^4-1^ Plackett-Burman fractional factorial design shown in Table 1 and described later, while control runs at pH 5.0 (Control A) and 6.0 (Control B) as well as runs to prepare the inoculum were carried out without polydextrose. The initial pH of the medium was adjusted to the selected values by addition of 1.3 N HCl.

To prepare the inoculum, aliquots of a cell suspension stored at −70 °C in 20% glycerol were added to 250 mL Erlenmeyer flasks containing 50 mL of MRS broth and shaken for 12 h at 100 rpm and 30 °C. After this period, 2 mL of cell suspension were taken to determine biomass concentration by optical density measurements at 600 nm (OD_600nm_) as described later. When the OD_600nm_ reached 0.8–0.9, which corresponds to 10^7^–10^8^ CFU mL^−1^, the inoculum suspension was transferred to 450 mL of the same MRS broth in 2 L Erlenmeyer flasks. Bacterial growth was followed for 24 h by triplicate sampling every 3 h.

*Enterococcus faecium* 101 and *Escherichia coli* ATCC 25922, stored at −70 °C in 20% glycerol, were used as the indicator strains to assay BLIS antimicrobial activity. For this purpose, they were grown overnight in Brain Heart Infusion (BHI) broth and MacConkey broth (Difco, Detroit, MI, USA), respectively. After that, 100 μL of suspensions of both strains were added to tubes containing 5.0 mL of the above broths and allowed to grow in a BOD incubator, model TE424 (Tecnal, Piracicaba, SP, Brazil), for 12–14 h at 30 °C. Ten microliters of suspension was periodically drawn to test bacteriocin antimicrobial activity.

### 2.2. Experimental Design

A 2^4-1^ Plackett–Burman fractional factorial design was used to study the effects of four independent variables, namely pH, polydextrose concentration, rotational speed, and temperature, on *P. pentosaceus* ATCC 43200 cultures. For this purpose, cell concentration and productivity after 24 h, maximum specific growth rate, specific rate of substrate (glucose) consumption, volumetric and specific lactic acid productivities, and yields of biomass and lactic acid on consumed substrate were selected as the responses. Such a design was composed of Controls A and B without polydextrose, 8 factorial runs, and 4 central point runs carried out in triplicate under the conditions defined in Table 1.

### 2.3. Analytical Procedures

Cell growth of *P. pentosaceus* ATCC 43200 was followed by optical density measurements at 600 nm (OD_600nm_) every 3 h for a total fermentation period of 24 h. The calibration curve *y* = 1.3677 *x* − 0.0008 (R^2^ = 0.9998) was constructed plotting data of cell mass concentration versus OD_600nm_ of cell suspensions at different dilutions (5×, 6×, 7×, 8×, 9×, 30×), and the results were expressed in g dry weight per liter (g_DW_ L^−1^).

The progressive pH decrease during fermentation was measured by a pH meter, model 400M1 (Quimis, Diadema, SP, Brazil).

Glucose and lactic acid concentrations were determined in triplicate samples of cell-free supernatants (CFS) by High-Performance Liquid Chromatography (HPLC), using standard solutions at concentrations in range to 0.1–2.0 g L^−1^ prepared with high-purity water (Milli-Q system, Millipore, Billerica, MA, USA). For this purpose, fermented broth samples were centrifuged at 5000× *g* for 20 min at 4 °C (U-32R Boeckel centrifuge, Hamburg, Germany) and then filtered through membranes with pore diameters of 45 µm (Merck-Millipore, Darmstadt, Germany). The CFS was analyzed by an HPLC system, model Prominence (Shimadzu, Kyoto, Japan), constituted by SIL-20ACHT auto-injector, UV SPD-20A light source, RI-10A refractive index detector, CBM-20A system controller, two LC20AD pumps, CTO20AC column oven, Aminex HPX-87H (300 × 7.8 mm, Bio Rad, Richmond, CA, USA) ion exclusion column and DGU-A20 degasser. The equipment was programmed to analyze 30 µL samples in the wavelength range from 190 to 230 nm, employing 5.0 mM sulfuric acid as a mobile liquid phase at 0.6 mL min^−1^ flowrate.

### 2.4. Determination of BLIS Antimicrobial Activity

Halo formation was considered to determine BLIS activity by the diffusion agar method against *E. faecium* 101 and *E. coli* ATCC 25922 as the indicator strains. Samples of CFS prepared by centrifugation at 5000× *g* for 10 min at 4 °C and filtration through a 45-μm porous membrane had their pH adjusted to 6.5 by the addition of 1.0 N NaOH and then were heated at 80 °C for 3 min to avoid bacteriocin inactivation by proteases. Quantitative analyses were carried out by the spot-on-lawn assay using suspensions of *E. faecium* 101 and *E. coli* ATCC 25922 at concentrations of 3 × 10^6^ CFU/mL and 6 × 10^6^ CFU/mL, respectively. Briefly, 10 μL of indicator strain suspension were spread onto 10 mL of BHI Agar (for *E. faecium*) or MacConkey Agar (for *E. coli*) previously supplemented with 1.0% agar. After medium solidification, 10 μL of each double dilution of every CFS were placed on the medium surface. Plates were prepared in duplicate and were incubated for 16–18 h at 37 °C. The antimicrobial activity of each CFS was expressed as arbitrary units (AU) per mL by the equation [35]:(1)$$\mathrm{Antimicrobial}\;\mathrm{activity}\;(\mathrm{AU}\;{\mathrm{mL}}^{-1})=\frac{D^{n}}{P}\;\times \;1000$$
where *D* is the dilution factor, *n* the first dilution exhibiting a negligible inhibition zone, and *P* the volume of sample deposited onto the agar surface.

### 2.5. Calculation of Fermentation Parameters

The maximum specific growth rate (*µ*_max_) and the generation time (*t_g_*) were calculated by the equations:(2)$$\mu_{\max}=\frac{1}{t_{f}-t_{0}}\;\ln\;\frac{X_{f}}{X_{0}}$$
(3)$$t_{g}=\frac{\ln2}{\mu_{\max}}$$
where *X*_0_ and *X_f_* are the biomass concentrations and the beginning and the end of the exponential phase, and *t*_0_ and *t_f_* the corresponding times.

The yields of biomass (*Y*_X/S_) and lactic acid (*Y*_P/S_) on consumed substrate and of lactic acid on biomass (*Y*_P/X_) after 24 h were calculated by the equations:(4)$$Y_{X/S}=\frac{X-X_{i}}{S_{i}-S}$$
(5)$$Y_{P/S}=\frac{P}{S_{i}-S}$$
(6)$$Y_{P/X}=\frac{P}{X-X_{i}}$$
where *P* is the lactic acid concentration after 24 h, *S_i_* and *S* are the initial substrate (dextrose) concentration and substrate concentration after 24 h, while *X_i_* and *X* are the corresponding values of biomass concentration.

Volumetric biomass (*Q*_X_) and lactic acid (*Q*_P_) productivity after 24 h were defined as:(7)$$Q_{X}=\frac{X-X_{i}}{t}$$
(8)$$Q_{P}=\frac{P}{t}$$
where *t* is the time of 24 h.

Specific substrate consumption rate (*q*_S_) and specific lactic acid productivity (*q*_P_) were calculated by dividing the corresponding volumetric values by the biomass concentration detected at the end of each run.

### 2.6. Statistical Study

Principal Component Analysis (PCA) was performed on the results of cell growth, pH, and lactic acid production after 24 h, as well as those of the BLIS concentration at the beginning and the end of logarithmic growth phase and 24 h. Response Surface Methodology (RSM) and Multiple Regression Analysis (MRA) were used to explain the responses obtained by PCA as functions of the selected independent variables [36].

## 3. Results and Discussion

### 3.1. Cell Growth

Figure 1 shows that in central point runs of the 2^4-1^ Plackett–Burman fractional factorial design, carried out with 1.0% polydextrose addition, *P. pentosaceus* ATCC 43200 reached a biomass concentration (4.75 ± 0.17 g_DW_ L^−1^) after 12 h about 25% higher than in Control A performed at the same starting pH (5.0) but only 8% higher than in the Control B (pH 6.0), after which it stopped growing. This result provides a first indication not only of the stimulating effect of polydextrose on *P. pentosaceus* growth during cultivation in MRS medium, but also the prominent role played by the medium acidity. However, it can be seen from the kinetic growth parameters listed in Table 2 that the maximum specific growth rate (*μ*_max_) followed an opposite trend, being in the central point runs (0.87 ± 0.01 h^−1^) about 7 and 2% lower than in Controls A and B, respectively.

Considering that this kinetic parameter is characteristic of the exponential phase of growth, the slower cell growth observed in central point runs may be due to the cell energy spent in this phase to cleave such a polysaccharide to dextrose, in order to make its transport through the cell membrane possible [6,37].

Extending the comparison to all the runs of the experimental design, one can see in Table 2 and Figure 2 that maximum values of biomass concentration after 24 h (*X* = 10.24 ± 0.16 g_DW_ L^−1^) and volumetric cell productivity (*Q*_X_ = 0.42 ± 0.01 g L^−1^ h^−1^) were obtained in run 8 carried out at highest levels of the independent variables (pH 6.0, 150 rpm, 35 °C) including polydextrose concentration (1.5%). Compared to Control B and central point runs (runs 9–12), these results represent 1.34- and 1.15-fold gains in terms of biomass concentration and 1.33- and 1.21-fold gains in terms of cell productivity.

These results corroborate what is described in the literature for *P. pentosaceus* growth in the pH range 5.0–8.0 [38,39,40]. To provide only a few examples, the strain *P. pentosaceus* Mees 1934 cultivated in MRS medium at pH 6.0 grew up to 6.5 g L^−1^, while at pH 6.5 it was unable to exceed 2.6 g L^−1^ [26]. *P. pentosaceus* CFR SIII isolated from cucumber was able to grow up in MRS at pH 6.5 and 37 °C for 16 h producing a heat-stable, anti-listerial and cell lytic bacteriocin [41]. However, there are studies that proved the ability of this bacterium to grow under more acidic conditions. For instance, *P. pentosaceus* ALP57 was able to grow effectively for 25 h on modified APT broth even at pH 4.1 [42].

The growth profile and bacteriocin production by *P. pentosaceus* are susceptible to changes in temperature because they happen simultaneously. According to Papagianni and Anastasiadou [39], the range of temperature from 28 to 35 °C could be considered appropriate for bacterial growth. Instead, the effect of rotational speed is not well described in the literature. Using a different approach, the shaking effect on the growth of *P. pentosaceus* Mees 1934 and *P. acidilactici* NRRL5627 was investigated by Anastasiadou et al. [26] and Anastasiadou et al. [38], who proposed a rotational speed of 150 rpm in MRS broth at pH = 6 and an oxygen level corresponding to 60% of its saturation in water as optimal growth conditions in a bioreactor. Under these conditions, bacteriocin was produced during early growth and until the stationary phase of growth.

Azevedo et al. [31] reported the improvement in the growth and bacteriocin-like inhibitory substance (BLIS) production by *P. pentosaceus* ATCC 43200 when it was cultivated anaerobically in MRS supplemented with 1.5% peptone in bioreactor for 24 h at 30 °C and agitation of 200 rpm. Under such optimal conditions, the cell mass concentration (3.41 g_DW_ L^−1^) was 66% higher, the generation time (1.28 h) 38% shorter and the BLIS activity against different indicator strains significantly higher than in MRS medium without any supplement taken as a control; it was in the range of 11.0–19.5 mm and the exponential phase started 4 h before. Production of BLIS by this *Pediococcus* strain was also influenced by the initial pH of the MRS medium as well as the addition of sugars (sucrose) and prebiotic (inulin). The CFS of the fermented control medium at pH 5.0 displayed antimicrobial activity against *E. faecium* 101 5.3% higher than that at pH 6.0, and even 20% higher than those of all supplemented media, regardless of the concentration of supplements. BLIS production was favored either at pH 5.0 or in the absence of any additional supplements, which were able, instead, to stimulate growth and lactate production by *P. pentosaceus* ATCC 43200 [32].

### 3.2. Polydextrose/Glucose Consumption and Lactic Acid Production

As is known, glucose is often the preferable carbon source of acid lactic bacteria when they are submitted to fermentation conditions, under which pyruvate is reduced to lactic acid by lactate dehydrogenase [43,44]. Particularly, *P. pentosaceus* ATCC 43200 was characterized as a homofermentative strain able to survive the anaerobic environment [38,44].

Polydextrose is a branched polymer, consisting of glucose monomers linked mainly via α- and β-1-6 glycosidic bonds, which is associated with the regulation of ATP-dependent transporter activities and participates in phosphoenolpyruvate production in the glycolytic pathway [35]. Lahtinen et al. [45] discovered that polydextrose was not completely consumed and fermented, and cells that cleaved it consumed glucose more quickly than fructose. Whereas polydextrose consumption by single bacterial cultures is not very well described [8], when the performance of bacterial blending was studied, the different strains seemed more suitable to ferment this polymer [7,46,47].

To shed further light on these topics, we assumed a delayed consumption of glucose in media supplemented with polydextrose due to a greater difficulty of the microorganism in acquiring the assimilable carbon source from the medium. Glucose preference as a carbon source was confirmed by the highest glucose consumption (15.00 ± 1.34 g L^−1^) found in run 2, performed in MRS broth supplemented with 0.5% polydextrose, which corresponded to 68.74% of the initial glucose and the highest specific rate of its consumption (*q*_S_ = 0.09 ± 0.02 g_S_ g_X_^−1^ h^−1^) with the exception of Control B (Table 3). These conditions favored fermentation, allowing a maximum lactic acid concentration (11.65 ± 0.47 g L^−1^) 47.47 and 25.32% higher than those obtained in Control A (6.12 ± 0.33 g L^−1^) and the central point runs (8.70 ± 0.50 g L^−1^) (Table 3 and Figure 3). As the yield of biomass on consumed substrate (*Y*_X/S_ = 0.46 ± 0.08 g_X_ g_S_^−1^) (Table 2) was 35.21% lower than the one of lactic acid on consumed substrate (*Y*_P/S_ = 0.71 ± 0.04 g_P_ g_S_^−1^) (Table 3), it is evident that cell growth, despite being satisfactory (*X* = 7.00 ± 1.11 g L^−1^, *µ*_max_ = 0.87 ± 0.01 h^−1^, *t_g_* = 0.83 ± 0.01 h) (Table 2), was the secondary carbon source destination in the bacterium under these conditions.

Lower *q*_S_ values were obtained when the broth was supplemented with higher levels of polydextrose, which appears to be a rule in fermentation processes when complex molecules are partially cleaved and utilized as carbon sources. Further analyzing the results of Table 2 and Table 3, one can see that the highest lactic acid productivity (*Q*_P_ = 0.36 ± 0.02 g L^−1^ h^−1^) and yield of lactic acid on biomass (*Y*_P/X_ = 1.87 ± 0.14 g_P_ g_X_^−1^) were obtained in the central point runs, while the highest *Y*_P/S_ (1.35 ± 0.17 g_P_ g_S_^−1^) was in run 6; these were carried out using the intermediate and lowest concentrations of polydextrose, respectively, which confirms preference of this bacterial strain for a directly assimilable carbon source such as glucose from the beginning of growth.

### 3.3. Antimicrobial Activity of BLIS

*P. pentosaceus* ATCC 43200 CFS aliquots of fermented broths were withdrawn during the exponential phase of growth and tested for antimicrobial activity against two indicator strains, the one representative of gram-negative bacteria (*E. coli* ATCC 2592) and the other of gram-positive ones (*E. faecium* 101).

As shown in Figure 4, the antimicrobial activity of BLIS against *E. faecium* 101 was as high as 6400 AU mL^−1^ when tests were made applying samples from Control A, central point runs and run 5. This value was 4 to 8 times higher than those (800–1600 AU mL^−1^) obtained using fermented broths from other runs. Since both Control A and central point runs were performed under similar conditions except for the percentage of polydextrose supplemented, this result demonstrates that the presence of this polymer in the medium did not significantly influence BLIS production.

Moreover, the antimicrobial activity of CFS at the start of fermentation suggests that *P. pentosaceus* ATCC 43200 synthesized BLIS during its growth as a product of its primary metabolism, thereby confirming the results reported in the literature by Piva and Headon [29] and Azevedo et al. [48], and for other strains of the same species, namely *P. pentosaceus* BCC 3772 [49] and *P. pentosaceus* 05-10 [50]. Nonetheless, a comparison between the results of Figure 2 and Figure 4 reveals that maximum BLIS antimicrobial activity did not correspond to maximum biomass concentration in runs 4, 5 and 8, which suggests a possible mechanism of BLIS release in which some other released metabolite or even polydextrose is involved. However, further efforts would be needed to confirm this hypothesis.

Some authors observed that pediocin release from the cell to the broth occurred very well as soon as the broth underwent acidification. Abrams et al. [51] reported maximum antimicrobial activity of fermented Pk34 pediocin-containing MRS broth as high as 12,800 AU mL^−1^ against *Listeria monocytogenes* and 3200 AU mL^−1^ against *Enterococcus faecalis* ATCC 29212 at pH = 6. Papagianni and Sergeledis [52] described a 25% increase in pediocin PA-1 production by *P. pentosaceus* Mees 1934 (170 AU mL^−1^) by reducing the pH of MRS broth from 6.8 to 6.5. Finally, Engelhardt et al. [53] reported a pediocin production of 6400 AU mL^−1^ by *P. acidilactici* HA 6111-2 in acidic MRS (pH 3.5). 

The results obtained in the present work partially agree with the above findings, since the highest antimicrobial activity against *E. faecium* 101 (6400 AU mL^−1^) was detected in acidic (pH 5.0) media (Control A and central point runs); however, overly acidic conditions (pH 4.0) remarkably reduced this activity, likely due to possible partial inactivation of bacteriocin or BLIS or even interference with its mechanism of action. In this respect, it is worth remembering that pore formation happens quickly in the host cell as soon as the identified pediocin or BLIS is connected to cell receptors placed on the cell wall surface. As a harmful consequence, ion transport is highly affected, inducing ion concentration instability between the intra- and intercellular environments [54,55,56]. Bacteriocins such as pediocin D were shown to interrupt cell membrane synthesis because they have affinity to glycerolipid molecules [57]. Obviously, strong acidity may have interfered with any of these events.

Figure 4 also shows the results of the antimicrobial activity of fermented broths (CFS) against *E. coli* ATCC 25922 as a representative of gram-negative bacteria. It can be seen that maximum BLIS activity (6400 AU mL^−1^) occurred under less acidic conditions than against *E. faecium* 101, which leaves us leaning towards interference on the mechanism of action. Gram-negative bacteria are in fact scarcely susceptible to bacteriocin action because in order for the bacteriocin to penetrate them, it must also cross the external membrane [58,59]. To overcome this problem, Tiwari et al. [60] structured a hybrid bacteriocin using a derivative N-terminal portion from pediocin PA-1 and C-terminal portion from enterocin 50–52. On plates containing hybrid bacteriocin and pathogenic gram-negative bacteria, these authors could quantify minimum inhibitory concentrations 64 times lower than that obtained for pediocin PA-1. Cizeikiene et al. [61], who analyzed the formation of halos obtained by applying CFS of the broth fermented by *P. pentosaceus* KTU05-10 on agar plates containing *E. coli* ATCC 25922 and *E. coli* 110, observed a maximum bacteriocin concentration (6400 AU mL^−1^) coincident with the maximum one observed in run 2 in this study.

These results as a whole show how much the conditions of bacteriocins or BLIS productions affect their concentration, activity and even targets.

### 3.4. Statistical Analyses

Statistical analyses were carried out on the results pertaining to cell growth, pH decrease, substrate consumption and concentrations of products at the end of cultures. As a first step, Principal Component Analysis (PCA) was performed aiming to take advantage of the correlation among responses. Such a statistical tool is increasingly used also in biotechnology owing to several advantages, among which the reduction of multidimensional data sets to a lower number of dimensions for further analysis stands out [62]. Data variances calculated by Minitab were 24.88, 4.09 and 3.88 for components 1 (PC1), 2 (PC2) and 3 (PC3), respectively, and the cumulative variance was equivalent to 80.12% of total variability, corresponding to 62.22, 10.2 and 7.7%, respectively.

Microbial behavior could be understood through the coefficients of responses estimated by the software Minitab 17, for which we considered linear combinations of experimental data of cell, substrate (glucose), lactate and BLIS concentrations as well as pH. The software was able to estimate absolute values for each coefficient, which expressed how participative each result was. Values of all estimated coefficients that were part of the previously mentioned components are listed in Table 4.

Examining the first component (PC1), there were relevant values of coefficients in particular intervals of time, namely cell growth (9–18 h), substrate consumption (3–21 h), lactic acid production (6–12) and pH decrease (9–18 h). Gathering this information, we noticed that PC1 expressed exponential growth of *P. pentosaceus* ATCC 43200. As for the second component (PC2), initial biological results obtained between 3 and 4 h were the most significant; therefore, we assumed that it represented the lag phase of growth and was associated with initial BLIS production. Furthermore, the third component (PC3) was composed of maximum coefficients of BLIS activity when this peptide tested against *E. faecium* 101 (0.326) and *E. coli* ATCC 25922 (0.414).

Response Surface Methodology (RSM) and Multiple Regression Analysis (MRA) were used to explain the responses obtained from PCA as function of pH, polydextrose concentration, rotational speed, and temperature. As shown in Table 5, there was a model associated with each component.

For each component, interactions between the variables and their significance for biological responses were also evaluated. Significance was assumed when the *p* values were <0.05. pH and polydextrose concentration had statistically significant effects (*p* = 0.000) on both cell growth and BLIS production, which were evaluated according to their linear effects. Although three different rotational speeds (50, 100 and 150 rpm) were tested, this variable was not statistically significant (PC1: *p* = 0.460 and PC2: *p* = 0.620).

However, when agitation was combined with pH, their interaction strongly affected cell growth (*p* = 0.000). As can be seen in Figure 5, the conditions of run 4 led to an increase in microbial growth (high values of PC1). These conditions could be considered excellent for cell growth until 18 h, but cell concentration subsequently decreased to 7.08 ± 0.41 g L^−1^. Even though runs 2, 6 and 8 resulted in PC1 equivalent to run 5, run 8 showed the maximum cell concentration (*X*_max_ = 10.24 ± 0.16 g L^−1^).

In addition, as illustrated in Table 6, the combination of pH and temperature exerted statistically significant effects (*p* = 0.01) on both cell growth and BLIS production, while the most noticeable quadratic effect was that of pH (*p* = 0.00). The importance of this effect is confirmed in runs 1, 3, 5 and 7 whose initial pH was 4.0. Despite the combinations between pH and rotational speed or temperature, cell growth was impaired (*X* = 5.00 ± 0.23 g·L^−1^, 1.09 ± 0.34 g·L^−1^, 5.17 ± 0.28 g·L^−1^ and 5.25 ± 0.03 g·L^−1^ in runs 1,3, 5 and 7, respectively) by acidification of the culture medium at the start of fermentations, and their PC1 were smaller than −5. 3D. Response surface plots were then generated aiming to find the optimal conditions able to ensure the highest values of PC3, i.e., to maximize BLIS production. Figure 5 shows that run 2 provided the best result of antimicrobial activity against *E. coli* ATCC 25922 (6400 AU mL^−1^) and consequently allowed a PC3 equivalent to 3.880.

Therefore, we can infer that these conditions may be the optimal ones for BLIS production. Furthermore, the initial BLIS production explained by PC2 was the highest in central runs, when strong antimicrobial activity against *E. faecium* (3200 AU mL^−1^) was noticed. On the other hand, the low value of PC2 in run 4 is consistent with the slight antimicrobial activity against the same strain (800 AU mL^−1^). The effect of each variable/factor (pH, Polydextrose, Rotation speed, and Temperature) on the responses (PC1, PC2, and PC3) are presented in the Pareto charts and main effects plots provided in Figure 6.

Following these steps, 3D Response surface plots were generated, aiming to find the optimal conditions able to ensure the highest values of PC3, i.e., to maximize BLIS production. Figure 5 shows that run 2 provided the best result of antimicrobial activity against *E. coli* ATCC 25922 (6400 AU mL^−1^) and consequently allowed a PC3 equivalent to 3.880. Therefore, we can infer that these conditions may be the optimal ones for BLIS production. 

## 4. Conclusions

The effect of polydextrose concentration on *P. pentosaceus* ATCC 43200 fermentation in MRS broth has been investigated under different operating conditions according to a Plackett–Burman design, paying particular attention to cell growth and BLIS production by this strain. As for cell growth, a polydextrose concentration of 1.5% (run 8) almost tripled the maximum cell concentration (*X*_max_ = 10.24 ± 0.16 g L^−1^) compared to Control A at pH 5.0 (3.78 ± 0.05 g L^−1^) with a consequent increase in cell productivity (*Q*_X_ = 0.42 ± 0.01 g L^−1^ h^−1^), although the highest maximum specific growth rate (*µ*_max_ = 0.94 ± 0.01) was obtained in the absence of polydextrose (Control A). Instead, no direct influence of the polydextrose concentration on BLIS production was found. In fact, the BLIS concentration was almost the same in the fermented broths from the Control (MRS) and the central point runs (MRS + 1% polydextrose), when *E. faecium* 101 was used as a target microorganism for its quantification. Unexpectedly, relevant results were obtained in terms of antibacterial activity against the gram-negative strain *E. coli* ATCC 25922 of the fermented broth (CFS) from run 2 (6400 AU mL^−1^), which suggests that more in-depth studies in this regard should be carried out in the future. To complete this study, a statistical analysis of the results was finally proposed using the so-called Principal Component Analysis. Through three models, we were able to confirm that interactions of (i) pH and rotational speed or (ii) pH and temperature are more important for cell growth and BLIS production than those involving polydextrose concentration. In conclusion, the analysis of the response surface suggested the addition of polydextrose in a concentration of 0.5% (run 2) aiming to optimize the growth of *P. pentosaceus* and the simultaneous production of BLIS. Since the microbial activity of BLIS can be greatly influenced by its degree of purity, further studies will deal with BLIS purification.

## Figures and Tables

**Figure 1 microorganisms-10-01898-f001:**
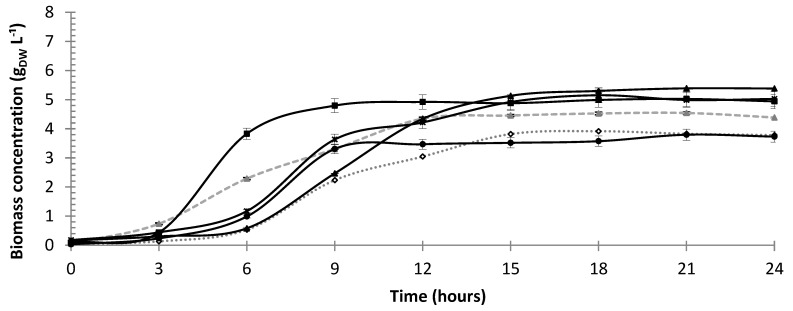
Growth of *P. pentosaceus* ATCC 43200 under different culture conditions. Central runs: 9 (■), 10 (●) 11 (▲) and 12 (✱). Control runs without polydextrose: A (◊); B: (▲). For conditions of runs and controls see Table 1.

**Figure 2 microorganisms-10-01898-f002:**
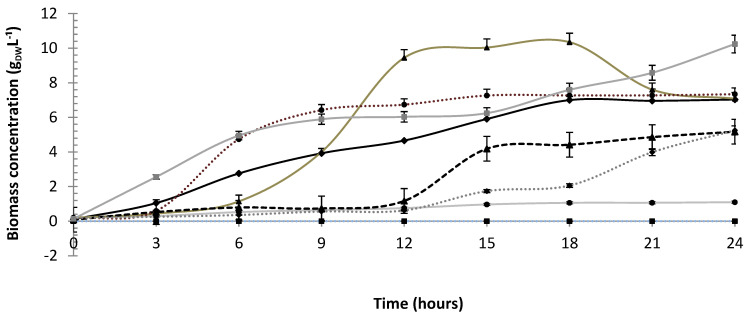
Growth of *P. pentosaceus* ATCC 43200 under different culture conditions. Runs: 1 (■); 2 (♦); 3 (●); 4 (▲); 5 (--▲--); 6(…●…); 7 (…♦…); 8 (■). For conditions of runs and controls see Table 1.

**Figure 3 microorganisms-10-01898-f003:**
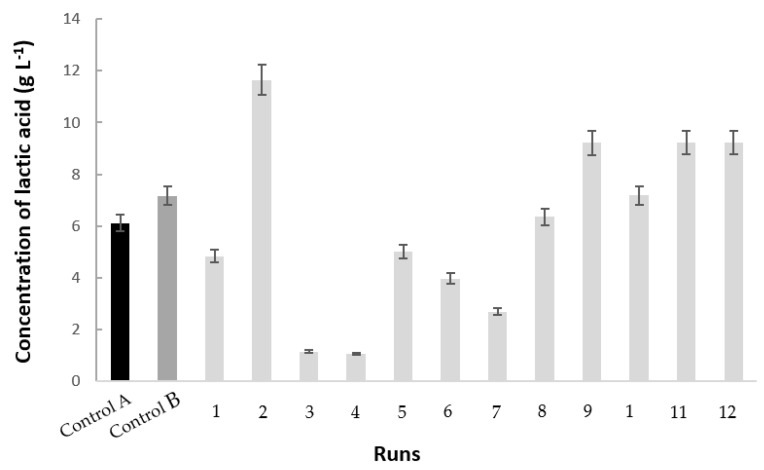
Lactic acid production by *P. pentosaceus* TCC 43200 during cultivations in MRS broth in for 24 h. A = Control A (■); B = Control B (■); runs 1-12 (■). For conditions of runs and controls see Table 1.

**Figure 4 microorganisms-10-01898-f004:**
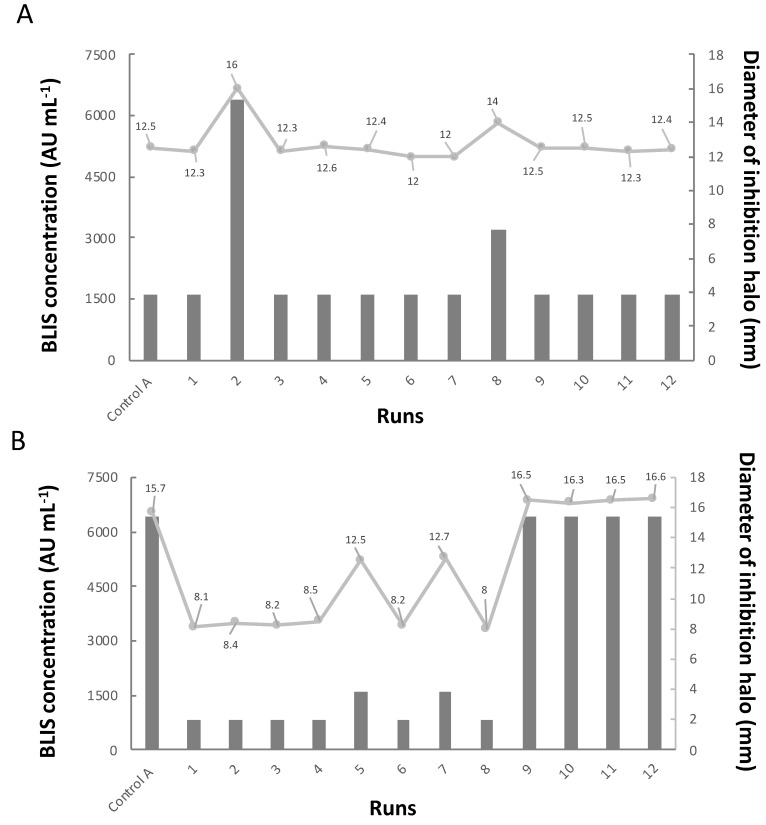
BLIS concentration expressed by AU mL^−1^ (■) and diameter of inhibition halo expressed by millimeter (mm) (■) of *P. pentosaceus* ATCC 43200 BLIS against *E. coli* ATCC 25922 (**A**) and *E. faecium* 101 (**B**) selected as Gram-negative and Gram-positive indicator strains.

**Figure 5 microorganisms-10-01898-f005:**
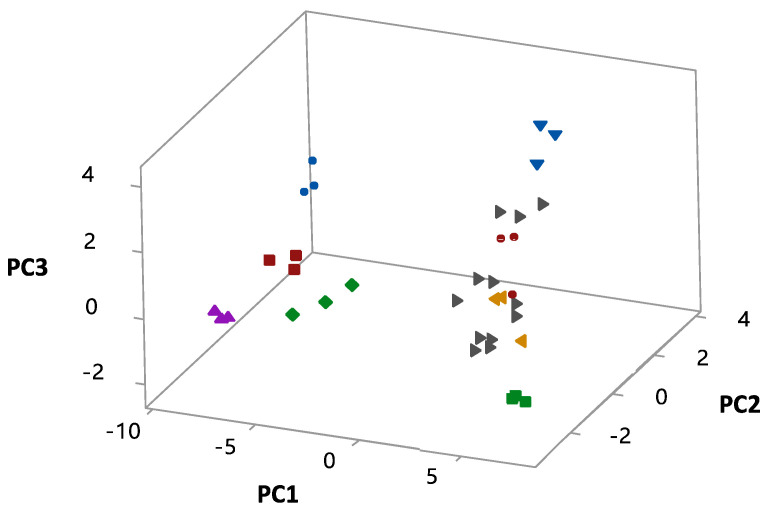
Growth conditions associated with biological answers illustrated by components 1 (exponential cell growth), 2 (cell growth in the lag phase) and 3 (BLIS production). Runs: (■) 1; (▼) 2; (▲) 3; (■) 4); (●) 5; (◄) 6; (⧫) 7; (●) 8; (►) Central runs.

**Figure 6 microorganisms-10-01898-f006:**
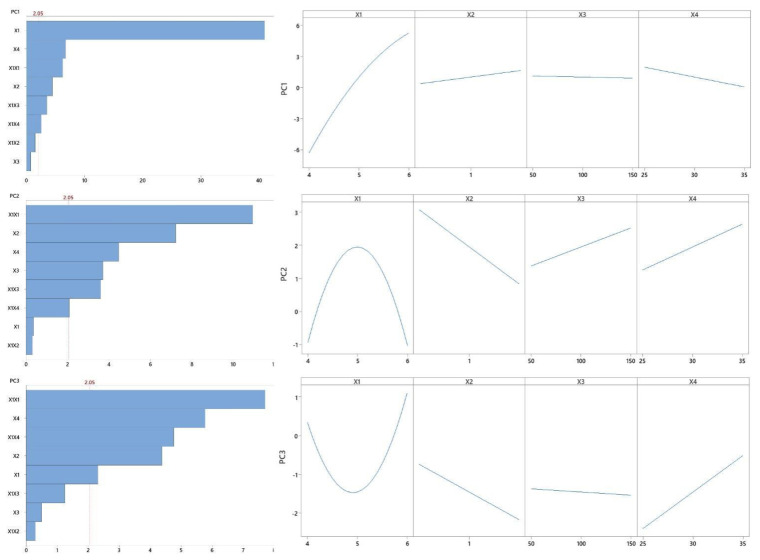
Pareto charts (**left**) and main effects plots (**right**) of the effect of the variables/factors (X1: pH, X2: Polydextrose, X3: Rotation speed, and X4: Temperature) on the responses (PC1, PC2, and PC3).

**Table 1 microorganisms-10-01898-t001:** 2^4-1^ Plackett-Burnman factorial design used to investigate the influence of pH, polydextrose concentration, rotational speed and temperature on *P. pentosaceus* ATCC 43200 cultivations.

Run	pH	Polydextrose Concentration (%)	Rotational Speed (rpm)	Temperature (°C)
Control A ^a^	5.0	0	100	30
Control B ^b^	6.0	0	100	30
1	4.0	0.50	50	25
2	6.0	0.50	50	35
3	4.0	1.50	50	35
4	6.0	1.50	50	25
5	4.0	0.50	150	35
6	6.0	0.50	150	25
7	4.0	1.50	150	25
8	6.0	1.50	150	35
9 ^c^	5.0	1.00	100	30
10 ^c^	5.0	1.00	100	30
11 ^c^	5.0	1.00	100	30
12 ^c^	5.0	1.00	100	30

^a^ Control at pH 5.0; ^b^ Control at pH 6.0. ^c^ Central point runs.

**Table 2 microorganisms-10-01898-t002:** Kinetic parameters and yields of *P. pentosaceus* ATCC 43200 cultivations.

Run	*X* (g_DW_ L^−1^) ^a^	*µ*_max_ (h^−1^) ^b^	*t*_g_ (h) ^c^	*Y*_X/S_ (g_X_ g_S_^−1^) ^d^	*Y*_P/X_ (g_P_ g_X_^−1^) ^e^	*Q*_X_ (g_X_ L^−1^ h^−1^) ^f^
Control A ^g^	3.78 ± 0.06	0.94 ± 0.01	0.75 ± 0.02	0.46 ± 0.00	1.63 ± 0.07	0.16 ± 0.00
Control B ^h^	4.38 ± 0.01	0.89 ± 0.03	0.89 ± 0.03	0.38 ± 0.02	1.65 ± 0.13	0.18 ± 0.00
1	5.00 ± 0.23	0.93 ± 0.02	0.76 ± 0.02	0.61 ± 0.15	1.02 ± 0.05	0.20 ± 0.01
2	7.00 ± 1.11	0.87 ± 0.01	0.83 ± 0.01	0.46 ± 0.08	1.56 ± 0.20	0.29 ± 0.04
3	1.09 ± 0.34	0.84 ± 0.16	0.88 ± 0.21	0.52 ± 0.22	1.52 ± 0.10	0.37 ± 0.01
4	7.08 ± 0.41	0.99 ± 0.01	0.70 ± 0.00	2.96 ± 0.34	0.15 ± 0.12	0.29 ± 0.02
5	5.17 ± 0.27	0.84 ± 0.06	0.87 ± 0.07	0.75 ± 0.18	0.98 ± 0.08	0.21 ± 0.01
6	7.34 ± 0.70	0.71 ± 0.04	1.03 ± 0.06	2.40 ± 0.42	0.55 ± 0.02	0.21 ± 0.17
7	5.25 ± 0.03	0.95 ± 0.02	0.75 ± 0.03	1.10 ± 0.51	0.53 ± 0.35	0.21 ± 0.00
8	10.24 ± 0.16	0.77 ± 0.04	0.95 ± 0.06	1.14 ± 0.14	0.63 ± 0.05	0.42 ± 0.01
9 ^i^	4.94 ± 0.26	0.68 ± 0.01	1.07 ± 0.02	0.61 ± 0.02	1.90 ± 0.12	0.20 ± 0.01
10 ^i^	3.69 ± 0.08	0.90 ± 0.02	0.80 ± 0.02	0.61 ± 0.25	1.94 ± 0.37	0.15 ± 0.00
11 ^i^	5.38 ± 0.13	0.97 ± 0.00	0.72 ± 0.00	0.66 ± 0.05	1.76 ± 0.01	0.22 ± 0.00
12 ^i^	5.02 ± 0.21	0.92 ± 0.03	0.77 ± 0.03	0.61 ± 0.06	1.89 ± 0.07	0.20 ± 0.01
Mean ^j^	4.75 ± 0.17	0.87 ± 0.01	0.84 ± 0.02	0.62 ± 0.09	1.87 ± 0.14	0.19 ± 0.01

^a^*X* = Biomass concentration after 24 h; ^b^
*µ*_max_ = Maximum specific growth rate; ^c^
*t*_g_ = Generation time; ^d^
*Y*_X/S_ = yield of biomass on consumed substrate after 24 h; ^e^
*Y*_P/X_ = Yield of lactic acid on biomass after 24 h; ^f^
*Q*_X_ = Volumetric biomass productivity after 24 h; ^g^ Control at pH 5.0; ^h^ Control at pH 6.0; ^i^ Central point runs; ^j^ Mean values of central point runs.

**Table 3 microorganisms-10-01898-t003:** Main experimental results of *P. pentosaceus* ATCC 43200 cultivations related to substrate consumption and lactic acid formation. Values were collected after 24 h of fermentation.

Run	pH Decrease	*S* (g_S_ L^−1^) ^a^	*P* (g_P_ L^−1^) ^b^	*q*_S_ (g_S_ g_X_^−1^ h^−1^) ^c^	*q*_P_ (g_P_ g_X_^−1^ h^−1^) ^d^	*Y*_P/S_ (g_P_ g_S_^−1^) ^e^	*Q*_P_ (g_P_ L^−1^ h^−1^) ^f^
Control A ^g^	1.07 ± 0.04	8.08 ± 0.08	6.12 ± 0.33	0.09 ± 0.00	0.07 ± 0.00	0.76 ± 0.05	0.26 ± 0.01
Control B ^h^	1.89 ± 0.03	11.44 ± 0.58	7.16 ± 0.58	0.11 ± 0.00	0.07 ± 0.01	0.64 ± 0.07	0.30 ± 0.02
1	0.39 ± 0.09	7.75 ± 2.18	4.84 ± 0.43	0.07 ± 0.02	0.04 ± 0.00	0.65 ± 0.15	0.20 ± 0.02
2	1.87 ± 0.02	15.00 ± 1.34	11.65 ± 0.47	0.09 ± 0.02	0.06 ± 0.01	0.71 ± 0.04	0.44 ± 0.02
3	0.21 ± 0.06	1.72 ± 0.20	1.14 ± 0.53	0.09 ± 0.03	0.06 ± 0.04	0.67 ± 0.35	0.048 ± 0.02
4	1.90 ± 0.06	2.32 ± 0.12	1.06 ± 0.84	0.01 ± 0.00	0.01 ± 0.00	0.46 ± 0.38	0.044 ± 0.04
5	0.64 ± 0.01	6.76 ± 1.67	5.01 ± 0.07	0.06 ± 0.06	0.04 ± 0.04	0.76 ± 0.13	0.21 ± 0.03
6	2.05 ± 0.03	2.98 ± 0.58	3.97 ± 0.45	0.01 ± 0.02	0.02 ± 0.02	1.35 ± 0.17	0.16 ± 0.02
7	0.30 ± 0.01	4.58 ± 1.77	2.69 ± 1.80	0.04 ± 0.01	0.02 ± 0.01	0.53 ± 0.03	0.11 ± 0.08
8	1.94 ± 0.01	8.82 ± 0.09	6.35 ± 0.50	0.04 ± 0.00	0.02 ± 0.00	0.73 ± 0.12	0.26 ± 0.02
9 ^i^	1.14 ± 0.03	7.90 ± 1.42	9.21 ± 0.28	0.01 ± 0.01	0.08 ± 0.00	0.59 ± 0.05	0.38 ± 0.01
10 ^i^	1.26 ± 0.02	6.06 ± 2.77	7.18 ± 1.33	0.07 ± 0.03	0.08 ± 0.01	1.27 ± 0.29	0.30 ± 0.06
11 ^i^	1.21 ± 0.11	7.90 ± 0.56	9.22 ± 0.21	0.06 ± 0.01	0.07 ± 0.00	1.17 ± 0.09	0.38 ± 0.01
12 ^i^	1.26 ± 0.04	7.93 ± 0.56	9.22 ± 0.21	0.07 ± 0.007	0.08 ± 0.003	1.15 ± 0.089	0.38 ± 0.009
Mean ^j^	1.22 ± 0.05	7.45 ± 1.33	8.70 ± 0.50	0.05 ± 0.016	0.08 ± 0.006	1.04 ± 0.131	0.36 ± 0.02

^a^*S* = Substrate concentration after 24 h; ^b^
*P* = Lactic acid concentration after 24 h; ^c^
*q*_S_ = Specific rate of substrate consumption; ^d^
*q*_P_ = specific lactic acid productivity; ^e^
*Y*_P/S_ = Yield of lactic acid on consumed substrate after 24 h; ^f^
*Q*_P_ = Volumetric lactic acid productivity after 24 h; ^g^ Control at pH 5.0; ^h^ Control at pH 6.0; ^i^ Central point runs; ^j^ Mean values of central point runs.

**Table 4 microorganisms-10-01898-t004:** Coefficients of Principal Component Analysis (PCA) applied to the three main components (PC1, PC2 and PC3) of *P. pentosaceus* fermentation in MRS broth.

Time (h)	0	3	4	6	9	12	15	18	21	24
	**Biomass Concentration**									
PC1	−0.020	0.088		0.122	0.170	0.198	0.171	0.172	0.166	0.140
PC2	−0.388	−0.061		0.060	0.077	−0.063	−0.055	−0.062	−0.033	−0.044
PC3	−0.073	0.302		0.230	0.054	−0.044	0.022	0.099	0.192	0.224
	**Substrate concentration**									
PC1	−0.103	−0.188		−0.199	−0.199	−0.199	−0.183	−0.181	−0.132	−0.177
PC2	0.198	0.022		−0.055	−0.071	−0.089	−0.122	−0.177	−0.122	−0.202
PC3	0.297	0.166		0.123	0.101	0.073	0.073	0.022	0.111	0.098
	**Lactic acid concentration**									
PC1	0.144	0.177		0.191	0.202	0.199	0.177	0.163	0.161	0.166
PC2	−0.270	−0.211		−0.033	0.024	0.095	0.200	0.223	0.233	0.254
PC3	−0.155	−0.077		−0.044	0.022	0.093	0.140	0.073	0.054	0.099
	**pH**									
PC1	0.193	0.194		0.199	0.185	0.173	0.170	0.170	0.176	0.161
PC2	−0.023	−0.033		−0.055	−0.022	−0.202	−0.199	−0.202	−0.207	−0.199
PC3	0.092	−0.041		−0.116	−0.116	−0.061	−0.077	−0.055	0.043	0.088
	**BLIS activity against *E. faecium* 101**									
PC1			−0.022							−0.044
PC2			0.221							−0.303
PC3			−0.064							0.326
	**BLIS activity against *E. coli***									
PC1			0.033							0.072
PC2			−0.163							0.022
PC3			0.422							0.414

PC1 = 1st Component associated with cell growth; PC2 = 2nd Component associated with cell adaptation during the lag phase; PC3 = 3rd Component associated with BLIS production.

**Table 5 microorganisms-10-01898-t005:** Statistical models of Principal Component Analysis (PCA) applied to the three main components (PC1, PC2 and PC3) of *P. pentosaceus* fermentation in MRS broth.

Response	Model	R^2^	R^2^ Adj	R^2^ Pred
PC1	PC1 = −53.21 + 19.40 *X*_1_ − 0.87 *X*_2_ + 0.055 *X*_3_ − 0.554 *X*_4_ − 1.532 *X*_1_^2^ + 0.433 *X*_1_*X*_2_ − 0.001 *X*_1_*X*_3_ + 0.073 *X*_1_*X*_4_	0.9852	0.9809	0.9794
PC2	PC2 = −70.32 + 28.46 *X*_1_ − 1.75 *X*_2_ + 0.077 *X*_3_ − 0.184 *X*_4_ − 2.932 *X*_1_^2^ − 0.101 *X*_1_*X*_2_ − 0.011 *X*_1_*X*_3_ + 0.076 *X*_1_*X*_4_	0.8919	0.8599	0.8536
PC3	PC3 = 67.87 − 25.57 *X*_1_ − 0.94 *X*_2_ + 0.022 *X*_3_ − 0.606 *X*_4_ + 2.188 *X*_1_^2^ − 0.101 *X*_1_*X*_2_ – 0.004 *X*_1_*X*_3_ + 0.166 *X*_1_*X*_4_	0.8402	0.7928	0.7582

PC1 = 1st Component associated with cell growth; PC2 = 2nd Component associated with cell adaptation during the lag phase; PC3 = 3rd Component associated with bacteriocin production. R^2^ = coefficient of determination; R^2^ adj = adjusted coefficient of determination; R^2^ pred = predicted coefficient of determination. *X*_1_ = pH; *X*_2_ = polydextrose concentration (%); *X*_3_ = rotational speed (rpm); *X*_4_ = temperature (°C).

**Table 6 microorganisms-10-01898-t006:** Analysis of variance for each component with linear and quadratic functions and interactions among pH, polydextrose concentration, rotational speed, and temperature.

	PC1			PC2			PC3		
Source	df	SS	*p*	df	SS	*p*	df	SS	*p*
Model	8	858.20	0.00	8	127.80	0.00	8	90.80	0.00
Linear	4	829.50	0.00	4	49.39	0.00	4	37.23	0.00
pH	1	797.90	0.00	1	0.08	0.72	1	3.43	0.03
Polydextrose	1	9.50	0.00	1	30.00	0.00	1	12.29	0.00
Rotational speed	1	0.61	0.47	1	7.87	0.00	1	0.16	0.62
Temperature	1	21.70	0.00	1	11.48	0.00	1	21.35	0.00
Quadratic	1	18.60	0.00	1	68.42	0.00	1	38.00	0.00
pH × pH	1	18.60	0.00	1	68.42	0.00	1	38.00	0.00
Interactions	3	10.20	0.00	3	10.01	0.00	3	15.57	0.00
pH × Polydextrose	1	1.09	0.14	1	0.06	0.76	1	0.06	0.77
pH × R. speed	1	5.89	0.00	1	7.45	0.00	1	1.00	0.22
pH × Temperature	1	3.17	0.02	1	2.50	0.05	1	14.51	0.00
Error	27	12.86		27	15.50		27	17.27	
Total	35	871.1		35	143.3		35	108.1	

Legend: df = degrees of freedom; SS = sum of squares; and *p* = significance (*p*-value).

## Data Availability

All data generated or analyzed during this study are included in this article.

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
