# Peer review of "Effect of Polydextrose on the Growth of Pediococcus pentosaceus as Well as Lactic Acid and Bacteriocin-like Inhibitory Substances (BLIS) Production"

_microorganisms, 2022, doi:10.3390/microorganisms10101898_

Round 1

Reviewer 1 Report

In the present manuscript, Maria Carolina W. Porto and co-workers described the effect of polydextrose on the growth of Pediococcus pentosaceus cumulated with lactic acid and bacteriocin-like inhibitory substances (BLIS) productions.

Unfortunately, the article lacks essential details regarding: the calibration curve that was used to convert the microbial OD to g/L (from 2.3 sub-chapter), and the concentration of the pathogens used for the antimicrobial activity.

Further, there are some confusing details regarding the antimicrobial activity, the way of how arbitrary units were calculated (taking in account that, in the article was not presented any bacteriocin purification method).

In the figure 1, the representation of the growth curves for 9, 10, 11, and 12 conditions was different, taking in account that the Pediococcus pentosaceus was grown in identical conditions (pH 5.0, 100 rpm, 30 °C and 1.0% polydextrose concentration). So, I think that the experiment started with different inoculum concentrations!

The authors expressed the bacterial cell concentration in g/L but, normally this unit is used for dry-weight biomass (information that is missing from the manuscript)!

The graph with the lactic acid production should be changed with a column chart.

It is not clear what exactly was represented in the figure 4! The antimicrobial activity could not be expressed in AU/mL! Also, the picture with the antimicrobial activity showing the inhibition zone diameters, was not provided.

In the discussion, there were made some comments regarding the results obtained for the antimicrobial activity and the literature. Unfortunately, these comparations were not correctly because there are different experiment conditions and the pediocin activity was tested after its purification (Simha et al., 2012).

For other comments, please see the manuscript!

Author Response

Reviewer 1

(Changes in the text to satisfy the suggestions of this reviewer are written in blue)

  1. Unfortunately, the article lacks essential details regarding: the calibration curve that was used to convert the microbial OD to g/L (from 2.3 sub-chapter), and the concentration of the pathogens used for the antimicrobial activity.

Answer: The equation and the information to obtain the calibration curve used to convert the microbial OD to g/L is reported for P. pentosaceus ATCC 43200 in lines 3-5 of Section 2.3. Cell concentrations of the suspensions of pathogens used for antimicrobial activity have been added in the text (Section 2.4, lines 7-8).

  1. Further, there are some confusing details regarding the antimicrobial activity, the way of how arbitrary units were calculated (taking in account that, in the article was not presented any bacteriocin purification method).

Answer: The way how arbitrary units were calculated is now explained better, showing the equation used (last 5 lines of Section 2.4). There is a debate in the literature about the meaning of the quantity expressed in AU/mL: Several authors consider it “antimicrobial activity” as in this paper (https://doi.org/10.1111/jam.13646, https://doi.org/10.1017/S0022029915000163), other authors, among which the reviewer, consider it “BLIS concentration” (https://doi.org/10.1016/j.lwt.2016.04.058, https://doi.org/10.1155/2021/7190152). Since the debate is still open, the authors preferred to maintain the former denomination, unless the reviewer believes absolutely necessary the latter option.

  1. In the figure 1, the representation of the growth curves for 9, 10, 11, and 12 conditions was different, taking in account that the Pediococcus pentosaceus was grown in identical conditions (pH 5.0, 100 rpm, 30 °C and 1.0% polydextrose concentration). So, I think that the experiment started with different inoculum concentrations!

Answer: The authors understand the perplexity of the reviewer about the variability noticed for four runs carried out under the same conditions, including the same inoculum concentration. However, a large variability is typical of any microbial culture, and especially of those of bacteriocin-producing strains, where the formation of the antimicrobial throughout the run unavoidably interferes with their normal metabolism. In other words, this additional factor of variability adds to the typical variability of normal cultures, leading often to rather different results among repetitions. This is the reason for which, when using an experimental design like the one adopted in this study, a reasonably large number of repetitions at the central point is required just to minimize the experimental error. In our case we used four repetitions (runs 9-12), but the variability remained rather high. However, we believe that it does not prejudice the overall significance of our results. 

  1. The authors expressed the bacterial cell concentration in g/L but, normally this unit is used for dry-weight biomass (information that is missing from the manuscript)!

Answer: The reviewer is completely right. The correct unit is g/L dry weight instead of simple g/L. This correction has been made throughout the whole text. In particular, this change was particularly important in the sentence following the equation of the calibration curve (Section 2.3, line 5).

  1. The graph with the lactic acid production should be changed with a column chart.

Answer: As suggested, the graph of the lactic acid production (Figure 3) has changed to a column chart.

  1. It is not clear what exactly was represented in the figure 4! The antimicrobial activity could not be expressed in AU/mL! Also, the picture with the antimicrobial activity showing the inhibition zone diameters, was not provided.

Answer: As explained in our answer to your previous question 2, there is a debate in the literature about the meaning of the quantity expressed in AU/mL: Several authors consider it “antimicrobial activity” as in this paper (https://doi.org/10.1111/jam.13646, https://doi.org/10.1017/S0022029915000163), other authors, among which the reviewer, consider it “BLIS concentration” (https://doi.org/10.1016/j.lwt.2016.04.058, https://doi.org/10.1155/2021/7190152). Since the debate is still open, the authors preferred to maintain the former denomination. We have added Equation (1) (last 5 lines of Section 2.4) which has been used in the past by many authors to calculate the antimicrobial activity just expressed in AU/mL.

In the discussion, there were made some comments regarding the results obtained for the antimicrobial activity and the literature. Unfortunately, these comparations were not correctly because there are different experiment conditions and the pediocin activity was tested after its purification (Simha et al., 2012).

Answer: The authors removed the reference to the work of Simha et al. (2012), but preferred not to remove the remaining discussion due to the relative paucity of articles in the specific subject of this paper (Section 3.3, second paragraph after Figure 4).

Further comments in the manuscript

Answer: The additional comments in the manuscript have been taken into consideration in this revision in accordance to the above responses.

Reviewer 2 Report

The manuscript entitled “Effect of polydextrose on the growth of Pediococcus pentosaceus as well as lactic acid and bacteriocin-like inhibitory substances (BLIS) productions” is a well-written research article with extensive work. The manuscript reflects the merit of the work and expertise of the researchers involved in this work. The manuscript is well collated with the results and outcomes of the research and well-articulated. However, a few basic comments are made to improve the article.

Manuscript can be accepted after minor revision

1.      Section 2.3 : pHmeter , use hyphen

2.      E. faecium is not Italicized on many occasions in the manuscript

3.      Units: uniformity in the units g.L-1 and g/L

4.      Kindly correct the highlighted texts in the caption and state, what do you mean by empty brackets Figure 4. Antimicrobial activity (AU/mL) of P. pentosaceus ATCC 43200 BLIS against E.a coli ATCC 25922 ( ) and E.s faecium 101 ( ) selected as Gram-negative and Gram-positive indicator strains

5.      Is it possible to provide the Response Surface Methodology plot in 3D surface illustration?

6.      Author mention the lack of literature, however, a few more recent studies can be referred:

·         Farinha, L. L., Sabrina S. Sabo, Maria C. Porto, Ellen C. Souza, Maricê N. Oliveira, and Ricardo PS Oliveira. "Influence of prebiotic ingredients on the growth kinetics and bacteriocin production of Lactococcus lactis." Chem. Eng 43 (2015): 313-318.

·         Hussien, Heba, Hagar Saeed Abd-Rabou, and Marwa A. Saad. "The Impact of Incorporating Lactobacillus Acidophilus Bacteriocin with Inulin and FOS on Yogurt Quality." (2022).

·         Juliyarsi, Indri, Yulianti Fitri Kurnia, Yudha Endra Pratama, and Dhiva Rezzy Pratama. "The quality of fermented goat milk produced by Pediococcus acidilactici BK01 on refrigerator temperature." Biodiversitas Journal of Biological Diversity 21, no. 10 (2020).

·         Bertuccioli, Alexander, Alfredo Bressan, Marco Neri, Diego Vergoni, and Giordano Zonzini. "Specific prebiotic composition for precision bacterial therapy in patients with irritable bowel syndrome." International Journal on Nutraceuticals, Functional Foods and Novel Foods (2022).

·         Danilčenko, Honorata. "Changes in the Quality of Inulin-Based Products During Storage." In Jerusalem Artichoke Food Science and Technology, pp. 213-227. Springer, Singapore, 2022.

·         Jagtiani, Ekta, and Sachin Adsare. "Microencapulsation: Probiotics, Prebiotics, and Nutraceuticals." (2022).

Author Response

Reviewer 2

(Changes in the text to satisfy the suggestions of this reviewer are written in green)

The manuscript entitled “Effect of polydextrose on the growth of Pediococcus pentosaceus as well as lactic acid and bacteriocin-like inhibitory substances (BLIS) productions” is a well-written research article with extensive work. The manuscript reflects the merit of the work and expertise of the researchers involved in this work. The manuscript is well collated with the results and outcomes of the research and well-articulated. However, a few basic comments are made to improve the article.

Manuscript can be accepted after minor revision

  1. Section 2.3: pHmeter, use hyphen

Answer: As suggested, the hyphen has been added in the word pH-meter (Section 2.3, 2nd paragraph, line 1).

  1. faecium is not Italicized on many occasions in the manuscript

Answer: Thanks. E. faecium has been italicized in the whole manuscript (example, Section 3.1, last paragraph, line 11).

  1. Units: uniformity in the units g.L-1 and g/L

Answer: As requested, the unit g/L has been replaced by g L-1 throughout the entire text. As a result, the same has been done for other units.

  1. Kindly correct the highlighted texts in the caption and state, what do you mean by empty brackets Figure 4. Antimicrobial activity (AU/mL) of  pentosaceusATCC 43200 BLIS against E.a coli ATCC 25922 ( ) and E.s faecium101 ( ) selected as Gram-negative and Gram-positive indicator strains

Answer: The authors are sorry for this mistake. The names of E. coli and E. faecium have been written correctly in the legend of Figure 4, and the symbols indicating the colors of columns have been added. Please, notice that this figure has been converted to a column chart to satisfy the suggestion of another reviewer.

  1. Is it possible to provide the Response Surface Methodology plot in 3D surface illustration?

Answer: Response Surface Methodology plots were obtained separately because of the large number (four) of independent variables; therefore, it was not possible to produce a 3D surface illustration.

  1. Author mention the lack of literature, however, a few more recent studies can be referred:
  • Farinha, L. L., Sabrina S. Sabo, Maria C. Porto, Ellen C. Souza, Maricê N. Oliveira, and Ricardo PS Oliveira. "Influence of prebiotic ingredients on the growth kinetics and bacteriocin production of Lactococcus lactis." Chem. Eng 43 (2015): 313-318.
  • Hussien, Heba, Hagar Saeed Abd-Rabou, and Marwa A. Saad. "The Impact of Incorporating Lactobacillus Acidophilus Bacteriocin with Inulin and FOS on Yogurt Quality." (2022).
  • Juliyarsi, Indri, Yulianti Fitri Kurnia, Yudha Endra Pratama, and Dhiva Rezzy Pratama. "The quality of fermented goat milk produced by Pediococcus acidilactici BK01 on refrigerator temperature." Biodiversitas Journal of Biological Diversity 21, no. 10 (2020).
  • Bertuccioli, Alexander, Alfredo Bressan, Marco Neri, Diego Vergoni, and Giordano Zonzini. "Specific prebiotic composition for precision bacterial therapy in patients with irritable bowel syndrome." Nutrafoods, 1 (2022): 331-336.
  • Danilčenko, Honorata. "Changes in the Quality of Inulin-Based Products During Storage." In Jerusalem Artichoke Food Science and Technology, pp. 213-227. Springer, Singapore, 2022.
  • Jagtiani, E., Adsare, S. Microencapsulation: Probiotics, Prebiotics, and Nutraceuticals. J. Nanotechnol. Nanomaterials. (2022) 3(1):34-60.

Answer: All the articles suggested by the reviewer have been cited in different parts of the Introduction (Introduction, 3rd paragraph, line 2; 4th paragraph, lines 3 and 4; last paragraph, lines 1 and 2).

Reviewer 3 Report

The present paper relates to the setup of an experimental design in order to highlight any possible effect of the tested parameter on cell growth and BLIS production.

The paper is well written and I think it can focus the interest of readers. There is a main point that in my opinion has to be analysed before paper acceptance. In page 9, authors stated that “P. pentosaceus ATCC 43200 CFS aliquots of fermented broths were withdrawn during the exponential phase (3-4 h) of growth and tested for antimicrobial activity against two indicator strains”. Looking at the time course of cell concentration in Figure 1 and 2, in some trials at the indicated time (3-4 h) cultures are not in the exponential phase. My doubt is that the BLIS activities reported are underestimated in some trials, and as consequence the obtained conclusions not correct.

Bacteriocins and in general BLIS as correctly written, are primary metabolites, thus associated to the cell growth. It seems strange that runs 4,5 and 8, highlighting the highest cell growth values, were not associated with the highest BLIS production.

Authors should provide data on bacteriocin production at the end of the exponential growth phase, that is the point in which the BLIS production is at its best. Only with those data it will be possible to compare the results.

Here follows a list with other revisions suggested:

-Page 2, third paragraph: “… Gram-positive bacteria such as Listeria monocytogenes and Listeria innocua, and their possible harmful effects to food consumers [23]”. L. innocua is not a pathogen, so it can be deleted from the sentence. In the same paragraph:  …Gram-positive bacteria such as Listeria ivanovii subsp. ivanovii ATCC 19119 (3.3 x 106 AU/mL) and Enterococcus faecium HKLHS (2.6 x 104 AU/mL).”. I do not understand the values in parenthesis: are these the MIC (Minimal Inhibitory Concentration? Or the maximum bacteriocin production? Pease explain.

- The final paragraph of the Introduction section is not written in proper English, I would suggest to rephrase it

- Page 4, paragraph 2.3: the last line reports 0.6 ml.min -1. Please delete the dot between mL and min.

- Table 2 should be presented in one page, and not cut into two pages. Please insert inside the Materials and methods section how the µmax and the other parameters were calculated.

- Page 6: there is no space between the legend of Table 2 and the subsequent text; please provide

- In Figure 1 and 2 and 3: I would suggest to insert in the legend only the number of the run of the trial as reported in Table 1 and not all the parameters applied (not pH 4.0, 150 rpm, 35 °C and 0.5% polydextrose concentration as example…, but use Run …).

- Page 6 last paragraph: for the citation  [33,34,35], I would prefer [33-35]. Also in page 10, there is [52,53,54], please revise as [52-54].

- page 7 first paragraph: “E. faecium 101”. The name should be written in italic.

- page 7, paragraph 3.2: “Lahtinen et al. (2010) [41] discovered that polydextrose was not completely consumed and fermented, and cells that cleaved it contained three units of cyclic glucose or fructose,…” . As it is written, it seems that cells contains 3 units of glucose… the term IT  should be referred to polydextrose. Sephrase the sentence.

- In the second paragraph of chapter 3.2 there are a lot of Unit of measures that need revision (g.L-1, gS/gX.h-1, …). Please revise throughout the manuscript.

- Table 3: the column related to qs and qp should me analogues with the others, avoiding too much space between the average and the standard deviation. How did you calculate the substrate consumption in trials with polidextrose? Do you have a method to calculate the residual polidextrose in cultures?

- page 9, the legend of figure 4 is incomplete.

- Page 9 paragraph 3.3: Authors stated that samples were taken at 3-4 h during exponential phase to detect BLIS. However, figure 2 and 3 highlights that in some run the strain at this incubation time is at the very beginning of the growth (as in trials 4 and 5), and maybe the BLIS concentration, as they behave as primary metabolites, is at the very beginning.

- page 10 the sentence “…because for the bacteriocin to be active within them has the difficult challenge of having to cross the extracellular wall around the membrane [56,57].” Is badly written; please revise. And also more: “To overcome this problem; Tiwari et al….”. Eliminate the semicolon ; and use the colon, .

- page 13: please revise table 6 as the part related to PC3  has a missing row. Inside the column of PC2 there is a number which is not aligned with the others.

References:

- journals names have to be written abbreviated as requested by the style of the journal

- 3: the journal id Foods (plural)

- 20: Enterococcus faecium has to be written in italic

- 26: Curr. Microbiol. written in italic

- 32: PLoS ONE in italic

- 53: Antonie Van Leeuwenhoek: should be abbreviated and in italic.

Author Response

Reviewer 3

(Changes in the text to satisfy the suggestions of this reviewer are highlighted in yellow)

  1. The paper is well written and I think it can focus the interest of readers. There is a main point that in my opinion has to be analysed before paper acceptance. In page 9, authors stated that “P. pentosaceus ATCC 43200 CFS aliquots of fermented broths were withdrawn during the exponential phase (3-4 h) of growth and tested for antimicrobial activity against two indicator strains”. Looking at the time course of cell concentration in Figure 1 and 2, in some trials at the indicated time (3-4 h) cultures are not in the exponential phase. My doubt is that the BLIS activities reported are underestimated in some trials, and as consequence the obtained conclusions not correct.

Answer: The authors totally agree with the reviewer. It was a mistake of the authors to mention 3-4 h of growth as the end of the exponential phase in all the runs. Indeed, as correctly observed by the reviewer, some of the runs did not reach the end of the exponential growth phase after this time. However, there was no subestimation of BLIS activity because the authors did the runs separately and, for each of them, they waited for the true end of the exponential phase to take these samples. This correction has been made in the text by deleting the wrong information (Section 2.6, line 3; Section 3.3, 1st paragraph, line 2). 

  1. Bacteriocins and in general BLIS as correctly written, are primary metabolites, thus associated to the cell growth. It seems strange that runs 4,5 and 8, highlighting the highest cell growth values, were not associated with the highest BLIS production.

Answer: This doubt of the reviewer was the same that the authors had. The authors believe that this BLIS, although associated to cell growth, may be released through a mechanism in which some other released metabolite or even polydextrose is involved, thus leading to a gap between the maximum concentrations of biomass and BLIS. However, further efforts would be needed to confirm this hypothesis. This concept has been added in the text (Section 3.3, 1st paragraph after Figure 4, lines 5-10).

  1. Authors should provide data on bacteriocin production at the end of the exponential growth phase, that is the point in which the BLIS production is at its best. Only with those data it will be possible to compare the results.

Answer: As explained in the answer to question 2 of this reviewer, we actually took samples at the end of the exponential phase of each runs. It was a mistake of the authors to mention 3-4 h of growth as the end of the exponential phase in all the runs. This mistake has been corrected.

  1. Here follows a list with other revisions suggested:

-Page 2, third paragraph: “… Gram-positive bacteria such as Listeria monocytogenes and Listeria innocua, and their possible harmful effects to food consumers [23]”. L. innocua is not a pathogen, so it can be deleted from the sentence. In the same paragraph:  …Gram-positive bacteria such as Listeria ivanovii subsp. ivanovii ATCC 19119 (3.3 x 106 AU/mL) and Enterococcus faecium HKLHS (2.6 x 104 AU/mL).”. I do not understand the values in parenthesis: are these the MIC (Minimal Inhibitory Concentration? Or the maximum bacteriocin production? Pease explain.

Answer: Many thanks for these suggestions. Listeria innocua has been deleted from the sentence (page 2, last paragraph, line 10). 3.3 x 106 AU/mL and 2.6 x 104 AU/mL were the antimicrobial activities expressed in arbitrary units reported in cited studies. However, since we did no report the corresponding values also for the other microorganisms mentioned in this sentence, because were expressed in different units, we have preferred to delete these values (page 2, lines 3-4 from the bottom).

- The final paragraph of the Introduction section is not written in proper English, I would suggest to rephrase it

Answer: The final paragraph of the Introduction section has been rewritten.

- Page 4, paragraph 2.3: the last line reports 0.6 ml.min -1. Please delete the dot between mL and min.

Answer: Thanks. This correction has been made throughout the whole manuscript also for other units.

- Table 2 should be presented in one page, and not cut into two pages. Please insert inside the Materials and methods section how the µmax and the other parameters were calculated.

Answer: Table 2 is now presented in only one page (page 6). Information on calculated fermentation parameters has been added in the new Section 2.5.

- Page 6: there is no space between the legend of Table 2 and the subsequent text; please provide

Answer: As suggested, a line has been added between the legend of Table 2 at the bottom and the subsequent text.

- In Figure 1 and 2 and 3: I would suggest to insert in the legend only the number of the run of the trial as reported in Table 1 and not all the parameters applied (not pH 4.0, 150 rpm, 35 °C and 0.5% polydextrose concentration as example…, but use Run …).

Answer: As suggested, indication of the conditions of each run has been removed in the legends of Figures 1, 2 and 3.

- Page 6 last paragraph: for the citation  [33,34,35], I would prefer [33-35]. Also in page 10, there is [52,53,54], please revise as [52-54].

Answer: This change has been made in the text (2nd line after Figure 2; Section 3.3, 3rd paragraph after Figure 4, line 9).

- page 7 first paragraph: “E. faecium 101”. The name should be written in italic.

Answer: This change has been made in the text (Section 3.1, last paragraph, line 5 from the bottom).

- page 7, paragraph 3.2: “Lahtinen et al. (2010) [41] discovered that polydextrose was not completely consumed and fermented, and cells that cleaved it contained three units of cyclic glucose or fructose,…” . As it is written, it seems that cells contains 3 units of glucose… the term IT  should be referred to polydextrose. Rephrase the sentence.

Answer: Many thanks. The sentence has been simplified to “Lahtinen et al. [47] discovered that polydextrose was not completely consumed and fermented, and cells that cleaved it consumed glucose more quickly than fructose.” (Section 3.2, 2nd paragraph, lines 4-6).

- In the second paragraph of chapter 3.2 there are a lot of Unit of measures that need revision (g.L-1, gS/gX.h-1, …). Please revise throughout the manuscript.

Answer: As suggested, all these units have been revised following the criterion: g L-1, gS gX-1 h-1, etc., throughout the whole manuscript (examples Section 3.2, 3rd paragraph, lines 7, 11 and 12).

- Table 3: the column related to qs and qp should me analogues with the others, avoiding too much space between the average and the standard deviation. How did you calculate the substrate consumption in trials with polidextrose? Do you have a method to calculate the residual polidextrose in cultures?

Answer: As suggested, columns with qs and qp values have been formatted removing spaces. As is now specified in the Materials and Methods section, we intended as “substrate” dextrose contained in the MRS broth (20 g/L), rather than polydextrose, which we supposed to act as a prebiotic (with negligible consumption). Details on the calculation of fermentation parameters have been added in the new Section 2.5.

- page 9, the legend of figure 4 is incomplete.

Answer: The symbols lacking in the legend of this figure have been added.

- Page 9 paragraph 3.3: Authors stated that samples were taken at 3-4 h during exponential phase to detect BLIS. However, figure 2 and 3 highlights that in some run the strain at this incubation time is at the very beginning of the growth (as in trials 4 and 5), and maybe the BLIS concentration, as they behave as primary metabolites, is at the very beginning.

Answer: As explained in our answer to the above question 2, it was a mistake of the authors to mention 3-4 h of growth as the end of the exponential phase in all the runs. Indeed, as correctly observed by the reviewer, some of the runs did not reach the end of the exponential growth phase after this time. However, there was no subestimation of BLIS activity because the authors did the runs separately and, for each of them, they waited for the true end of the exponential phase to take these samples. This correction has been made in the text by deleting the wrong information (Section 2.6, 1st paragraph, line 3; Section 3.3, 1st paragraph, line 2).

- page 10 the sentence “…because for the bacteriocin to be active within them has the difficult challenge of having to cross the extracellular wall around the membrane [56,57].” Is badly written; please revise. And also more: “To overcome this problem; Tiwari et al….”. Eliminate the semicolon ; and use the colon, .

Answer: Thanks. The first sentence has been improved (Section 3.3, penultimate paragraph, lines 5-7), and the semicolon in the second has been replaced by a comma (Section 3.3, penultimate paragraph, line 7).

- page 13: please revise table 6 as the part related to PC3  has a missing row. Inside the column of PC2 there is a number which is not aligned with the others.

Answer: Thanks. In Table 6, the missing raw has been added in PC3 column, and the number has been aligned with others in the PC2 one.

References:

- journals names have to be written abbreviated as requested by the style of the journal

- 3: the journal id Foods (plural)

- 20: Enterococcus faecium has to be written in italic

- 26: Curr. Microbiol. written in italic

- 32: PLoS ONE in italic

- 53: Antonie Van Leeuwenhoek: should be abbreviated and in italic.

Answer: All these changes have been done.

Reviewer 4 Report

The paper “Effect of polydextrose on the growth of Pediococcus pentosaceus as well as lactic acid and bacteriocin-like inhibitory sub-stances (BLIS) productions” addresses a topic worthy of investigation. Although the topic could be of interest for Microorganisms, the paper does not meet the standards to be published at least in its present form. There are some technical and methodological issues to be addressed. The most important are as follows:

A)    Statistic: the use of a DoE is useful if after that data modelling is performed through response surface and desirability approaches to point out the effect of each variable. Such method should be used for each parameter reported in table 3 and for BLIS.

B)    Data on figures 1 and 2 have a too low standard deviation; thus, my question is: did authors used technical replicates or independent batches (I mean the analyse were performed twice on the same sample or on two different samples prepared separately?). The use of independent samples is advisable.

Author Response

Reviewer 4

(Changes in the text to satisfy the suggestions of this reviewer are written in red)

The paper “Effect of polydextrose on the growth of Pediococcus pentosaceus as well as lactic acid and bacteriocin-like inhibitory sub-stances (BLIS) productions” addresses a topic worthy of investigation. Although the topic could be of interest for Microorganisms, the paper does not meet the standards to be published at least in its present form. There are some technical and methodological issues to be addressed. The most important are as follows:

  1. Statistic: the use of a DoE is useful if after that data modelling is performed through response surface and desirability approaches to point out the effect of each variable. Such method should be used for each parameter reported in Table 3 and for BLIS.

Answer: The authors agree with the reviewer about the importance of using statistical approaches for modeling data obtained following a Design of Experiments. However, other statistical approaches are equally effective to this purpose in addition to Response Surface and Desirability function. Among them, the Principal Component Analysis (PCA) adopted in this study (Section 3.4) is increasingly used also in the biotechnology field owing to several advantages, among which it stands out the reduction of multidimensional data sets to a lower number of dimensions for further analysis (U. Roessner, A. Nahid, B. Chapman, A. Hunter, M. Bellgard, 1.31 - Metabolomics – The Combination of Analytical Biochemistry, Biology, and Informatics, In Comprehensive Biotechnology (M. Moo-Young, Ed.), Third Edition, Vol. 1, 2011, p. 435-447, https://doi.org/10.1016/B978-0-444-64046-8.00027-6). A sentence about the advantage of using this statistical approach has been added in the text (Section 3.4, 1st paragraph, lines 4-7), as well as this reference. Further analysis was done in our study by Response Surface Methodology (RSM) and Multiple Regression Analysis (MRA) to explain the responses obtained from PCA as function of pH, polydextrose concentration, rotational speed, and temperature (Section 3.4, 2nd paragraph after Table 4, first 3 lines). Therefore, the authors believe that the statistical approach used, alternative to that suggested by the reviewer, is suitable for the purposes of this study.

  1. Data on figures 1 and 2 have a too low standard deviation; thus, my question is: did authors used technical replicates or independent batches (I mean the analyse were performed twice on the same sample or on two different samples prepared separately?). The use of independent samples is advisable.

Answer: The reviewer is right. It was an error of the authors, who used, at the moment of inserting the error bars in the graphs, the values referred to technical replicates of a batch instead of those referred to independent batchs. Nonetheless, the means were correct. Therefore, we have revised both figures inserting the correct error bars.

Round 2

Reviewer 1 Report

In the present manuscript, Maria Carolina W. Porto and co-workers described the effect of polydextrose on the growth of Pediococcus pentosaceus cumulated with lactic acid and bacteriocin-like inhibitory substances (BLIS) productions.

The manuscript was improved with essential information but I still consider that the antibacterial activity is significantly influenced by the purification process of the biological compound.

Further, the picture showing this antimicrobial activity was not provided. Furthermore, the AU/mL is the BLIS concentration that confer different  (higher, medium, minimum or absent) antimicrobial activities observed through inhibition zone diameter. 

So, from my point of view, in the figure 4 you should have on the OY axis the BLIS activity (AU/mL) and on the OX axis, the inhibition zone diameter expressed in millimeters.

The authors from the paper you highlighted as a response to the comment no. 6 (https://doi.org/10.1111/jam.13646), showed the antimicrobial activity of different purified proteins concentrations (ug/uL) measuring the diameter of the zone of inhibition (mm).

Concluding, I still consider that figure no 4 need to be improved, as I mentioned above.

Best regards,

Author Response

In the present manuscript, Maria Carolina W. Porto and co-workers described the effect of polydextrose on the growth of Pediococcus pentosaceus cumulated with lactic acid and bacteriocin-like inhibitory substances (BLIS) productions. The manuscript was improved with essential information but I still consider that the antibacterial activity is significantly influenced by the purification process of the biological compound.

Answer. The authors agree with the reviewer about the great influence of the purification process on the antibacterial activity. However, purification of BLIS will be one of the objectives of the next efforts (Conclusions, last two lines). Further, the picture showing this antimicrobial activity was not provided. Furthermore, the AU/mL is the BLIS concentration that confer different (higher, medium, minimum or absent) antimicrobial activities observed through inhibition zone diameter. So, from my point of view, in the figure 4 you should have on the OY axis the BLIS activity (AU/mL) and on the OX axis, the inhibition zone diameter expressed in millimeters. The authors from the paper you highlighted as a response to the comment no. 6 (https://doi.org/10.1111/jam.13646), showed the antimicrobial activity of different purified proteins concentrations (ug/uL) measuring the diameter of the zone of inhibition (mm).

Answer. We appreciate your comment. We specifically updated Figure 4 by adding the diameter of the inhibition zone in millimeters.

Reviewer 3 Report

Authors have applied the requested revision. In my opinion the paper can be accepted in the present form.

Author Response

Authors have applied the requested revision. In my opinion the paper can be accepted in the present form.

Answer. Thank you so much for considering the manuscript.

Reviewer 4 Report

My issue on statistic was not addressed; although the authors used a robust analysis, the effect of each variable/factor should be assessed.

Author Response

My issue on statistic was not addressed; although the authors used a robust analysis, the effect of each variable/factor should be assessed.

Answer. The ANOVA results provided in Table 6 allow one to assess the effect of each variable/factor (pH, Polydextrose, rotational speed, tempertature) on the responses (PC1, PC2, and PC3). The higher the SS values the higher the effect of the variable/factor on the response. These effects are also presented in the Pareto chart and main effects plots provided in Figure 6.